# Dysfunction of Prkcaa Links Social Behavior Defects with Disturbed Circadian Rhythm in Zebrafish

**DOI:** 10.3390/ijms24043849

**Published:** 2023-02-14

**Authors:** Han Hu, Yong Long, Guili Song, Shaoxiong Chen, Zhicheng Xu, Qing Li, Zhengli Wu

**Affiliations:** 1Key Laboratory of Freshwater Fish Reproduction and Development (Ministry of Education), College of Fisheries, Research Center of Fishery Resources and Environment, Southwest University, Chongqing 400715, China; 2State Key Laboratory of Freshwater Ecology and Biotechnology, Institute of Hydrobiology, Chinese Academy of Sciences, Wuhan 430072, China; 3College of Fisheries and Life Science, Dalian Ocean University, Dalian 116023, China

**Keywords:** zebrafish, *prkcaa*, social behavior, circadian rhythm, gene expression

## Abstract

Protein kinase Cα (PKCα/PRKCA) is a crucial regulator of circadian rhythm and is associated with human mental illnesses such as autism spectrum disorders and schizophrenia. However, the roles of PRKCA in modulating animal social behavior and the underlying mechanisms remain to be explored. Here we report the generation and characterization of *prkcaa*-deficient zebrafish (*Danio rerio*). The results of behavioral tests indicate that a deficiency in Prkcaa led to anxiety-like behavior and impaired social preference in zebrafish. RNA-sequencing analyses revealed the significant effects of the *prkcaa* mutation on the expression of the morning-preferring circadian genes. The representatives are the immediate early genes, including *egr2a*, *egr4*, *fosaa*, *fosab* and *npas4a*. The downregulation of these genes at night was attenuated by Prkcaa dysfunction. Consistently, the mutants demonstrated reversed day–night locomotor rhythm, which are more active at night than in the morning. Our data show the roles of PRKCA in regulating animal social interactions and link the social behavior defects with a disturbed circadian rhythm.

## 1. Introduction

Protein kinase C (PKC) represents a large protein kinase family, the members of which function as molecular machines to decode the second messengers Ca^2+^ and diacylglycerol (DAG) [1]. The PKCs are classified into three groups, namely conventional (cPKC), novel (nPKC) and atypical (aPKC) on the basis of their structure and requirements for cofactors [2,3]. The cPKCs (α, βI, βII and γ) require both Ca^2+^ and DAG for activation; the nPKCs (δ, ε, η, θ and μ) require only DAG; the aPKCs (ζ, ι, λ) require neither Ca^2+^ nor DAG but can be stimulated by phosphatidylserine [2,4]. All PKCs share the conserved architecture with an N-terminal regulatory region, a hinge region and a C-terminal catalytic region [3,5]. The regulatory domain of the PKCs contains a pseudosubstrate segment that has an autoinhibitory effect and serves as a key molecular switch in the regulation of the enzyme’s activity [5]. Upon binding to the second messengers or protein partners, the pseudosubstrate is released from the substrate-binding cavity, resulting in the activation of the kinase [5]. The activated PKCs phosphorylate many target proteins at the serine and threonine residues, thus translating various extracellular signals into crucial cellular processes.

PRKCA/PKCα is a member of the conventional PKCs. It was found to be extensively expressed and to play pivotal roles in multiple cellular processes, including cell proliferation, differentiation, adhesion, migration and apoptosis [6,7]. A gene-knockout mouse (*Mus musculus*) model has been generated to investigate the biological functions of the *Prkca* gene in vivo. Evidence obtained from gene-knockout mouse indicated that a deficiency in Prkca enhances insulin signaling through PI3K [8]. Prkca is an important modulator of bipolar cell signal transduction and termination; a deficiency in the *Prkca* gene impairs the visual processing of mouse [9]. Prkca is the critical signaling intermediator for T helper 17 (Th17) cell activation and plays important roles in Th17-cell-mediated autoimmunity; Prkca-deficient mice failed to mount appropriate Th17 cell immune responses [10].

PRKCA is involved in multiple human diseases, such as heart failure, cancer and arterial thrombosis; it is also regarded as a potential therapeutic target [7]. Furthermore, PRKCA modulates the light entrainment of the mammalian circadian clock [11]. Evidence from the gene-knockout mouse showed that PRKCA regulates PER2 (period circadian regulator 2) stability and subcellular localization through posttranslational modification [11]. PRKCA was revealed to be an integral part of the circadian feedback loop. It forms a complex with and phosphorylates the core circadian clock component, BMAL1; the depletion of endogenous PRKCA from the cultured cells shortened the circadian period [12]. PRKCA also participates in the epigenetic regulation of the circadian clock by phosphorylating LSD1 (lysine-specific demethylase 1), and the phosphorylated LSD1 forms a complex with CLOCK:BMAL1 to facilitate the transcriptional activity [13]. Consistent with the importance of PRKCA in regulating the circadian rhythm and the strong association between circadian rhythm disruption and mental health [14,15], the mutation of *Prkca* is strongly associated with human mental illnesses such as autism spectrum disorders (ASD) and schizophrenia [16,17,18]. ASD patients are characterized as having repetitive behavioral patterns and have problems with social communication and interaction [17]. Previous studies have suggested a function for PRKCA in regulating social behaviors; however, so far, there are no data to support this assumption.

Zebrafish (*Danio rerio*) are highly social and share genetic homology and behavioral patterns with humans [19,20]. It has become popular for studying the genetic or pharmacological mechanisms underlying complex social behavior and has been established as a prominent animal model for neurological disorders such as ASD [19,20,21,22]. Furthermore, zebrafish are amenable to genome editing, a promising tool for modeling human diseases [23]. For example, the zebrafish mutant model lacking the RAR-related orphan receptor A, paralog a (*roraa*) developed by using the clustered, regularly interspaced palindromic repeats (CRISPR)/Cas9 system successfully recapitulated the neuroanatomical features of ASD [24].

In this study, we generated *prkcaa*-knockout zebrafish lines and performed an array of behavioral assays to explore the roles and mechanisms of Prkcaa in regulating social behavior. We observed that a deficiency in Prkcaa resulted in anxiety-like behaviors, impaired social preference and shoaling behavior in zebrafish. RNA-sequencing assays revealed that Prkcaa dysfunction had a significant impact on the downregulation of a group of circadian genes at night. The results of locomotor activity analyses indicated that the *prkcaa*^−/−^ mutants are more active at night; namely, the mutation of *prkcaa* reversed the day–night locomotor rhythm of zebrafish. The *prkcaa*-knockout zebrafish successfully modeled the social disorder of humans, so it can serve as a valuable tool for drug screening. Our data offer in vivo evidence for the roles of PRKCA in regulating mental health and directly link impaired animal social behavior with a disturbed circadian rhythm. Furthermore, we identified immediate early genes *egr2a*, *egr4*, *fosaa*, *fosab* and *npas4a* as the candidates that are affected by the *prkcaa* mutation and that bridge animal social behavior and the circadian rhythm. These results provide novel insights into the molecular mechanisms underlying Prkcaa’s function.

## 2. Results

### 2.1. Molecular Characterization of Zebrafish Prkcaa and Prkcab

The *prkca* gene was duplicated in the genome of zebrafish, which led to the paralogs *prkcaa* and *prkcab*. The peptide sequences of these two paralogs are highly conserved, except for the N and C terminals and the linker sequence (Appendix A). The results of a phylogenetic analysis revealed that PRKCA/Prkca from mammals, chicken (*Gallus gallus*) and African clawed frog (*Xenopus laevis*) were clustered together. The clade of Prkcaa/Prkcab from zebrafish and medaka (*Oryzias latipes*) was clearly separated from that of the PRKCA/Prkca proteins. The clade of KPC2 from *Caenorhabditis elegans* and the PCK-53E from *Drosophila melanogaster* were located at the root of the phylogenetic tree (Figure 1A). The PRKCA/Prkca and Prkcaa/Prkcab from the representative vertebrates are highly conserved. The sequence identities among the proteins ranged from 84% to 94% (Figure 1B), suggesting a high level of functional similarity.

The transcriptional expression patterns of the zebrafish *prkcaa* and *prkcab* genes in the adult tissues were analyzed through qPCR assays. Both genes are actively expressed in the characterized tissues, except for the liver (Figure 1C). Although these two paralogs are highly conserved, their expression patterns considerably differ. The highest expression of *prkcaa* was found in the ovary, followed by the eye and the brain, while the highest transcription of *prkcab* was found in the brain, followed by the eye and the gill. The subcellular localization of zebrafish Prkcaa and Prkcab was also explored. Both proteins were evenly distributed in the cytoplasm when exotically expressed in the 293T cells. Furthermore, Prkcab was also found to be localized in the membrane (Figure 1D).

### 2.2. Generation of the prkcaa-Knockout Zebrafish Lines

We successfully generated knockout zebrafish lines for the *prkcaa* gene using the CRISPR/Cas9 system. The target sequence was localized in the seventh exon of the gene. Two mutant lines were generated and submitted to the China Zebrafish Resource Center (CZRC). The mutant lines were designated as ZKO3145A (3145A, −2 bp) and ZKO3145B (3145B, −11 bp) (Figure 2A). The genotypes of the mutant lines were confirmed by DNA sequencing. The mutations resulted in a frame shift and protein truncation. The truncated proteins were predicted to contain 250 and 247 amino acids, which were less than half of the full length and lack the catalytic domains. The mutants can develop and reproduce normally. No significant difference in body weight (Figure 2B), standard length (Figure 2C) or condition factor (Figure 2D) were found between the adults of the WT and 3145A mutants, indicating that a deficiency in Prkcaa had no effects on the growth or nutritional status of zebrafish.

### 2.3. Dysfunction of Prkcaa Led to Behavioral Defects in Zebrafish

Behavioral assays were performed to assess the effects of Prkcaa dysfunction on the behavior of zebrafish. The behavioral tests and the corresponding purposes are listed in Table 1.

#### 2.3.1. The *prkcaa*^−/−^ Mutants Exhibited Anxiety-Like Behavior

Novel tank tests were performed using the apparatus displayed in Figure 3A to investigate the novelty-evoked anxiety-like behaviors of the fish. As illustrated by the representative tracked trajectories (Figure 3B), the mutants demonstrated less exploratory activity into the top region of the tank than the WT counterparts. Quantitative analyses of the behavioral data indicated that the mutants had significantly higher total freezing time (*p* < 0.01, Figure 3C) and lower average velocity (*p* < 0.01, Figure 3D). The mutants demonstrated significantly higher latency to enter the top area (the time spent before the first entrance into the top, *p* < 0.001, Figure 3E). Furthermore, the mutants had a significantly lower number of entries into the top (*p* < 0.001, Figure 3F), time spent in the top (*p* < 0.001, Figure 3G) and distance traveled in the top (*p* < 0.001, Figure 3H). These end points are robust indicators for anxiety symptoms in zebrafish [25,26]. Therefore, the dysfunction of Prkcaa resulted in anxiety-like behavior in zebrafish.

#### 2.3.2. The *prkcaa*^−/−^ Mutants Displayed Impaired Aggressive Behavior

Mirror biting tests were performed, as illustrated in Figure 4A, to explore the effects of the *prkcaa* mutation on the social/aggressive behavior of zebrafish. The 3145A mutants demonstrated a significantly higher total freezing time (*p* < 0.001, Figure 4B), a lower average velocity (*p* < 0.01, Figure 4C) and a lower total distance traveled in the experimental tank (*p* < 0.001, Figure 4D) than the WT counterparts, upon the mirror image stimulation. The mutants also had a significantly lower number of entries into the approach area (AA, *p* < 0.01, Figure 4E), spent less time in the AA (*p* < 0.01, Figure 4F) and traveled less within the AA (*p* < 0.05, Figure 4G). Furthermore, the mutants demonstrated decreased mirror biting activity compared with the WT, which had a significantly lower mirror biting frequency (number of entries into the contact area, *p* < 0.001, Figure 4H) and mirror biting duration (*p* < 0.01, Figure 4I). These data indicate that the dysfunction of Prkcaa impaired the social/aggressive behavior of zebrafish.

#### 2.3.3. The *prkcaa*^−/−^ Mutants Demonstrated Decreased Social Preference

Social preference tests were performed, as illustrated in Figure 5A, to investigate the effects of the *prkcaa* mutation on the preference of zebrafish to conspecifics. The trajectories of the representative WT and mutant fish are shown in Figure 5B. The trajectory of the WT fish was concentrated in the conspecific sector (CS), while that of the 3145A mutant was evenly distributed between the CS and the empty sector (ES). Consistently, statistical analyses of the behavioral data indicated that the mutants had a significantly lower social preference value (*p* < 0.001, Figure 5C), ratio of time spent in the CS (*p* < 0.001, Figure 5D) and ratio of distance traveled in the CS (*p* < 0.05, Figure 5E) than the WT. Taken together, the dysfunction of Prkcaa led to the decreased social preference of zebrafish.

#### 2.3.4. The *prkcaa*^−/−^ Mutants Had Defects in Shoaling Behavior

To analyze shoaling behavior of the fish, individuals of the same genotype (WT and 3145A) were tested in groups (five fish per group), as illustrated in Figure 6A. Video was recorded and subjected to behavioral analysis for each group. The *prkcaa*^−/−^ mutants had a significantly higher average interfish distance (*p* < 0.001, Figure 6B), average nearest neighbor distance (*p* < 0.001, Figure 6C), shoal area (*p* < 0.001, Figure 6D and Appendix A) and frequency of excursion from the shoal (*p* < 0.001, Figure 6E and Appendix A) in comparison with the WT counterparts, suggesting that the mutants tend to form looser shoals. Furthermore, the dysfunction of Prkcaa significantly decreased the shoal polarization of zebrafish (*p* < 0.05, Figure 6F), indicating that the mutants are less cohesive in their swimming direction. These results demonstrate that Prkcaa plays important roles in the shoaling behavior of zebrafish.

### 2.4. Effects of prkcaa Mutation on Gene Expression

Gene expression in the brain of the WT and 3145A mutants were profiled by RNA-seq to explore the mechanisms underlying the roles of Prkcaa in regulating fish behavior. Given the effects of circadian rhythm on fish behavior, the fish were sampled both in the morning (9:00 a.m.) and at night (9:00 p.m.). In total, 16 libraries were generated and sequenced (four biological replicates for each treatment). From 19.00 to 27.72 M read pairs were generated for the libraries. About 99% of the raw reads past the read filtering threshold and more than 90% of the clean reads could be mapped to the reference genome of zebrafish (Appendix A). Gene transcriptional abundance (FPKM, fragments per kilobase per million mapped fragments, Appendix A) was calculated, and 21,823 genes were regarded as expressed (FPKM > 0.1 in all the samples of at least one experimental group).

The results of a principal component analysis (PCA) revealed that both the genotype of the fish (WT vs. 3145A) and the sampling time (night vs. morning) had significant effects on gene expression (Figure 7A). The first two PCs explained 40% (PC1) and 25% (PC2) of the variance in gene expression, respectively. Differentially expressed genes (DEGs) between the experimental groups were identified, and they are listed in Appendix A. The expressions of genes such as *fosab* (v-fos FBJ murine osteosarcoma viral oncogene homolog Ab), *gpr186* (G protein-coupled receptor 186) and *npas4a* (neuronal PAS domain protein 4a) (Appendix A) were analyzed by qPCR to validate the data of RNA-seq. The result of a Pearson correlation analysis revealed a significant correlation between the RNA-seq and qPCR data sets (R^2^ = 0.8171, *p* < 0.001, Appendix A). The numbers for the DEGs are displayed in Figure 7B. The 3145A mutants had 722 upregulated (U1) and 1624 downregulated genes (D1) in comparison with the WT in the morning (M: 3145A vs. WT). Similar numbers for DEGs (U2 and D2) were identified between the night samples (N: 3145A vs. WT). Furthermore, 1409 circadian genes were identified in the brain of the WT fish (WT: N vs. M), including 484 upregulated (U3) and 925 downregulated genes (D3). Considerably smaller numbers of circadian genes were identified for the 3145A mutants (3145A: N vs. M), which had 394 upregulated (U4) and 717 downregulated genes (D4).

The results of Venn analyses revealed that 175 circadian genes were affected by the *prkcaa* mutation (Figure 7C), including 44 night-preferring genes (demonstrating higher expression at night, GS1) and 131 morning-preferring genes (demonstrating higher expression in the morning, GS2). The Prkcaa-deficiency-affected circadian genes are listed in Appendix A. The results of the G-test indicated that the mutation of *prkcaa* had a significant effect on expression of the morning-preferring genes (*p* < 0.001) but had no significant effect on the night-preferring genes (*p* = 0.1045). The genes differentially expressed between WT_N and WT_M are displayed in the volcano plot (Figure 7D). The representative *prkcaa*-mutation-affected circadian genes, such as *npas4a*, *fosab* and *gpr186*, were indicated. These genes demonstrated lower expression at night in comparison with in the morning, and their downregulation at night was attenuated by Prkcaa dysfunction.

### 2.5. Functional Enrichments for the Differentially Expressed Genes

GO and KEGG pathway enrichment analyses were performed to explore the biological functions associated with the DEGs. Full lists of the enriched GO terms, the false discovery rate and the number and ratio of the associated genes are displayed in Appendix A. The term lists were simplified by REVIGO [27], and the terms with the lowest dispensability and highest uniqueness are displayed in Figure 8A. Biological processes such as photoperiodism, the circadian regulation of gene expression and the regulation of the circadian rhythm were enriched for the DEGs between the morning and night samples for both the WT and 3145A mutants. Chaperone-mediated protein folding and the organic hydroxy compound biosynthetic process were enriched only for the DEGs between WT_N and WT_M. Multiple immune-related processes such as the regulation of immune system process, leukocyte activation and leukocyte cell–cell adhesion were enriched for the genes differentially expressed between 3145A and WT, both in the morning and at night. MHC protein complex assembly and the execution phase of apoptosis were enriched for the DEGs between 3145A_M and WT_M, while biological processes such as leukocyte proliferation and plasma membrane invagination were affected by the *prkcaa* mutation only at night (3145A_N and WT_N).

The KEGG pathways enriched for the DEGs are listed in Appendix A. The circadian genes identified for the WT (WT_N vs. WT_M) and 3145A (3145A_N vs. 3145A_M) samples had different KEGG pathway enrichments. Pathways such as neuroactive ligand-receptor interaction and steroid biosynthesis were exclusively identified for WT, while Arginine biosynthesis was specific for the 3145A mutants (Figure 8B). The dysfunction of Prkcaa had different effects on the pathway enrichments in the morning and at night. Pathways such as herpes simplex virus 1 infection and Toll-like receptor signaling were enriched only for the DEGs between 3145A_M and WT_M, while phagosome and cell adhesion molecules were specific for the DEGs between 3145A_N and WT_N. These results provide an overview for the biological functions of zebrafish Prkcaa.

### 2.6. The Mutation of prkcaa Affected Genes Involved in Neural Activities

The alterations of fish behavior are intimately related to structural and functional changes in the nervous system [28,29]. To investigate the effects of the *prkcaa* mutation on the expression of genes involved in neural processes, the full GO term list for the DEGs between the 3145A_M and WT_M samples were checked to identify the terms associated with processes such as nervous system development, neuron differentiation and calcium signaling. The term list was simplified, and the nonredundant terms are displayed in Figure 9A. Terms including nervous system development, brain development, neuron differentiation and the nervous system process were significantly overrepresented (*p* < 0.01, Appendix A). Furthermore, there were also large numbers of DEGs associated with neural processes such as synaptic signaling, neuron projection extension and calcium-mediated signaling (Appendix A).

In total, 182 genes associated with neural processes were found to be affected by Prkcaa dysfunction and were classified into five clusters through K-means clustering (Figure 9B). Most of these genes were assigned to the first two clusters and demonstrated no circadian rhythm in expression. The mutation of *prkcaa* inhibited the cluster 1 genes, while it enhanced those of cluster 2. The genes of clusters 3 and 5 had a higher expression level in the morning; the mutation of *prkcaa* decreased their expression and abolished the morning–night rhythm. Moreover, the dysfunction of *prkcaa* enhanced the cluster 4 genes in the morning but had no effect on their expression at night. Coexpression networks for the genes of cluster 1 and cluster 2 were constructed using WGCNA. Genes such as *mob1a* (MOB kinase activator 1A), *coro1a* (coronin, actin binding protein, 1A) and *rhogd* (ras homolog gene family, member Gd) are the hubs of cluster 1 (Figure 9C); while *dpysl5b* (dihydropyrimidinase-like 5b), *pank2* (pantothenate kinase 2) and *igf2bp2a* (insulin-like growth factor 2 mRNA binding protein 2a) are the hubs of cluster 2 (Figure 9D). These genes are the typical downstream targets affected by the *prkcaa* mutation.

Together, the above data indicate that except for influencing the circadian genes, the mutation of *prkcaa* had considerable effects on the expression of the genes associated with nervous system development and neural activities. The altered expression of these genes may account for the abnormal behavior of the mutants.

### 2.7. The 3145A Mutants Had Inverse Rhythm of Locomotion Activity

Because the downregulation of a subset of circadian genes at night was attenuated in the 3145A mutants (Figure 7C,D), we hypothesize that the mutation of *prkcaa* may disturb the circadian rhythm of zebrafish. We first determined the expression of the representative circadian genes, including *fosaa* (v-fos FBJ murine osteosarcoma viral oncogene homolog Aa), *fosab* and *npas4a*, through qPCR (Figure 10A–C). These genes are established markers for neural activity [30,31,32]. The qPCR results indicated that the expression levels of *fosab* and *npas4a* were significantly higher (*p* < 0.05) in the 3145A_N than the WT_N samples, while a level of higher but not significant (*p* = 0.07) expression was found for *fosaa* in the 3145A_N samples.

Locomotion activity assays revealed that the WT fish demonstrated decreased locomotor activity at night, as expected; however, the 3145A mutants were more active at night than in the morning (Figure 10D). The WT fish swam longer in the morning than at night; on the contrary, the mutants swam longer at night (Figure 10E). The trajectories of the fish were divided into large, moderate, and small movements and freezing. The WT fish had a significantly higher ratio of large movement in the morning and a higher ratio of moderate movement at night; however, the mutants had more small movement in the morning and more moderate movement at night (Figure 10F). These results indicated that the 3145A mutants had inverse rhythms of locomotion activity, they were more active at night.

## 3. Discussion

PRKCA is listed as a strong (score 2) candidate causal gene for ASD by SFARI GENE (https://gene.sfari.org, accessed on 20 December 2022), a database that is centered on genes implicated in autism susceptibility. However, the in vivo role of PRKCA in regulating animal social behavior remains to be verified at an organism level. Here, we report on the generation and characterization of the *prkcaa*-deficient zebrafish model. The dysfunction of Prkcaa, the zebrafish ortholog of human PRKCA/PKCα, had no significant impact on the development, growth and reproduction of the fish. The results of the behavioral assays revealed that the mutation of *prkcaa* led to an anxiety-like manifestation and impaired social behavior in zebrafish. In comparison with the WT counterparts, the mutants were less aggressive, had a reduced preference for conspecifics and showed defects in shoaling behavior. Zebrafish are highly social and prefer to swim in cohesive shoals. The movement of fish in shoals has multiple advantages in foraging and avoiding predation [26]. Therefore, except for offering an extrapolation model for understanding functions of the human PRKCA in regulating mental health, Prkcaa may play critical roles in facilitating the adaptation of wild populations to the challenging environments with limiting food resources.

Because animal social behavior is determined mainly by brain functions, we explored the effects of the *prkcaa* mutation on gene expression in the brain both in the morning and at night. The RNA-seq data revealed an array of genes that demonstrated differential expression in the morning and at night. These genes were significantly enriched in circadian-related processes, as expected. However, different from the previous results that a disruption in PRKCA’s activity leads to altered expression of the core clock genes [13], none of the core clock genes, including *clocka* (clock circadian regulator a), *clockb*, *bmal1a* (basic helix-loop-helix ARNT-like 1a) and *bmal1b*, were found to be affected by Prkcaa dysfunction. The mutation of *prkcaa* also had no impact on the expression of the period circadian clock and the cryptochrome circadian regulator gene family members. Except for *prkcaa*, zebrafish have another *PRKCA* paralog, *prkcab*. It is possible that the functions of these two paralogs have markedly differentiated during evolution and that Prkcab is specialized to regulate the core circadian genes in zebrafish. Furthermore, given the high sequence identity between Prkcaa and Prkcab and the enhanced expression of *prkcab* in the *prkcaa*-deficient zebrafish (Appendix A), it is likely that the functions of Prkcaa may be partially compensated by Prkcab in the mutants.

Consistent with the functions of PRKCA in regulating the immune process [33], the dysfunction of Prkcaa led to the downregulation of multiple genes associated with the immune pathways. Interestingly, pathways such as Toll-like receptor signaling and NOD-like receptor signaling were exclusively enriched among the DEGs identified from the morning samples, while phagosome was overrepresented by the night DEGs (Figure 8B). It was reported that the immune system shows regularly recurring and rhythmic variations, and both the humoral and the cellular immune system function in a rhythmic manner [34,35]. The immune responses of Nile tilapia (*Oreochromis niloticus*) to lipopolysaccharide (LPS) were found to be gated by the time of day [36]. Furthermore, a couple of GO terms involved in neural functions were also significantly enriched for the genes affected by the loss of *prkcaa*. Together, our data shed further light on the multiplicity of PRKCA’s molecular functions and bridge the activities of PRKCA in circadian rhythm with processes such as immune and neural functions.

Although Prkcaa deficiency had no effect on the core clock genes, a significant impact was identified for the morning-preferring genes. The downregulation of 131 circadian genes at night was attenuated by the *prkcaa* mutation (Appendix A). A couple of light-induced immediate early genes (IEGs) were contained in this Prkcaa dysfunction-affected gene list, such as *egr2a* (early growth response 2a), *egr4*, *fosaa*, *fosab*, *fosb* and *npas4a*. These genes encode transcription factors, demonstrate day–night expression rhythms and play pivotal roles in regulating activity rhythms. Among the mammalian homologs of these interested transcription factors, *Egr2* is crucial for normal hindbrain development, and *Egr2*-null mice die perinatally [37]. EGR2 is involved in cognitive function and was reported to be a possible susceptibility gene for human bipolar disorder [38]. Both the transcript and protein expressions of *c-fos* (homolog of *fosaa* and *fosab*) are rhythmic [39,40]. Evidence from *c-fos*-null mice indicated that it was required for the normal entrainment of the biological clock [41]. Furthermore, there were intimate relationships among c-Fos expression, animal social status and circadian phase. For instance, individual housing increased c-Fos levels (contrast to those housed in groups of four) only during the dark phase of circadian cycle [42]. The *Npas4* gene (mammalian homolog of *npas4a*) is involved in orchestrating the transcriptional responses of SCN (suprachiasmatic nucleus) neurons; a deficiency in NPAS4 alters the circadian rhythms of mice [43]. *Npas4* was selectively induced by neuronal activity and was reported to gate cocaine-induced hyperlocomotion in mice [44]. Therefore, the deregulation of these circadian transcription factors may account for the altered social behavior of the mutants.

Another question that needs to be touched on is the molecular mechanisms whereby PRKCA orchestrates the rhythmic expression of the circadian transcription factors. Although mammalian PRKCA affects the expression of circadian genes by modulating the core clock regulators, the loss of *prkcaa* had no impact on the expression of the core clock genes in zebrafish. Moreover, the active Prkcab may partially compensate for the functions of Prkcaa. Therefore, the observed effects of the *prkcaa* mutation on the expression of the circadian gene are not likely to be caused by changes in the core clocks. The induction of *npas4a* in mouse neurons and human iPSC-derived brain organoids by synaptic stimuli is dependent on calcium signaling and is irrelevant with the kinase pathways, including PKA (protein kinase A) and MAPK (mitogen-activated protein kinase) cascades [44]. The calcium-signaling elements, including NMDA (*N*-methyl-d-aspartate) receptors, voltage-dependent calcium channels, nuclear calcium, CaMKII/IV (Ca^2+^/calmodulin-dependent protein kinase II/IV) and calcineurin determine the induction of *Npas4* [44]. On the other hand, PRKCA was reported to be a fundamental regulator of Ca^2+^ handing in myocytes, which alters sarcoplasmic reticulum Ca^2+^ loading and Ca^2+^ transient by directly phosphorylating protein phosphatase inhibitor-1 (I-1), changing the activity of protein phosphatase-1 (PP-1) and dephosphorylation of phospholamban (PLB, inhibitory protein of SERCA-2) [45]. Therefore, it is possible that Prkcaa modulates the expression of the circadian genes by influencing calcium signaling in the neural tissues.

## 4. Materials and Methods

### 4.1. Zebrafish Maintenance and Measurement

Zebrafish of AB strain was maintained in aquariums supplied with circulating water as previously described [46]. The water temperature was controlled at 28 ± 1 °C. The fish room was illuminated from 8:00 a.m. to 8:00 p.m. The fish were fed with nauplii of brine shrimp to satiation twice daily. Artificial reproduction of zebrafish was performed following the previous protocols [46]. The embryos and early larvae were incubated in E3 medium (5 mM NaCl, 0.17 mM KCl, 0.33 mM CaCl_2_, and 0.33 mM MgSO_4_, pH 7.2) at 28 °C using biochemical incubators obtained from Shanghai Jinghong laboratory instrument Co., Ltd. (Shanghai, China). To assess the growth and nutritional status of the fish, body weight was determined using an analytical balance (accuracy 0.01 g) from Mettler Toledo (Columbus, OH, USA); standard length (from the tip of the lower jaw to the posterior end of the hypural bone) was measured using a ruler; and condition factor was calculated as 100 × (body weight/standard length^3^).

### 4.2. Phylogenetic Analysis

Phylogenetic analysis was performed to elucidate the evolutionary relationship among the PRKCA proteins of multiple animal species. Sequences of PRKCA (or Prkcaa and Prkcab) from species including Homo sapiens, Macaca mulatta, Mus musculus, Gallus gallus, Xenopus laevis, Cyprinus carpio, Caenorhabditis elegans, Drosophila melanogaster and Danio rerio were downloaded from the GenBank database. The accession numbers of the protein sequences are listed in Appendix A. Complete sequence alignment was performed using clustalx (v2.1) [47], and a phylogenetic tree for the PRKCA proteins from multiple species was generated by MEGA (v11) [48] by using the maximum-likelihood method with default parameters.

### 4.3. Total RNA Extraction

Total RNA extraction was performed using TRIzol from Thermo Fisher (Waltham, MA, USA) according to the manufacturer’s instructions. Four-month-old adult zebrafish were used for tissue sampling. The fish were anesthetized with 160 mg/L of MS-222 (Sigma, St. Louis, MO, USA) before dissection. To characterize tissue expression patterns of the prkcaa and prkcab genes, tissues including brain, eye, gill, heart, intestine, kidney, liver, muscle, ovary and testis were collected from 3–5 wild-type zebrafish. The samples collected from the fish were mixed and subjected to total RNA extraction. Quality of the total RNA samples was checked by agarose electrophoresis. RNA concentration was measured using a Quawell Q5000 UV-Vis Spectrophotometer (Sunnyvale, CA, USA).

### 4.4. Real-Time Quantitative PCR Assay

Real-time quantitative PCR (qPCR) was performed using a Bio-Rad CFX Duet Real-Time PCR System (Hercules, CA, USA) to determine the level of gene transcription. The reagents, protocols, amplification program and method for data analysis were the same as previously described [49]. The 18s rRNA was used as the internal control to normalize expression of the prkcaa and prkcab genes among the adult tissues. Expression of the genes associated with nervous system development and neural activity was normalized to that of the actb1 gene. The sequences, amplicon size and efficiency of the primers used for qPCR assays are listed in Appendix A.

### 4.5. Molecular Cloning

Total RNA extracted from 96 hpf zebrafish larvae was subjected to first-stranded cDNA synthesis using the RevertAid First Strand cDNA Synthesis Kit (Thermo Fisher, Waltham, MA, USA). The coding sequences of zebrafish Prkcaa and Prkcbb were amplified using the primers prkcaa_cds-F/prkcaa_cds-R (CTCAAGCTTGCCACCATGGCTGATACACAAAGCAACGAGC/GGTGGATCCGCTTCAGTGTTGACAAGGGAGGGA; the underlined letters are restriction enzyme sites) and prkcab_cds-F/prkcab_cds-R (GATCTCGAGGCCACCATGGCTGATCATCTGATACAGATCA/GGTGGATCCGCTACCACACTGTGAATAGACGC), respectively. The amplified fragments were purified and inserted into the HindIII/BamHI and XhoI/BamHI sites of the pAcGFP-N1 vector (Clontech, CA, USA), respectively. The resulted fusion protein-expressing constructs pAc-Prkcaa-GFP and pAc-Prkcab-GFP were used to transform Trelief 5α-competent cells (TSINGKE, Beijing, China). Single colonies were picked, amplified and subjected to plasmid extraction. The cloned sequences were confirmed by DNA sequencing by Sangon Biotech (Shanghai, China).

### 4.6. Cell Culture, Transfection and Microphotography

Cell culture, transfection and microphotography were performed as previously described [50]. Briefly, the 293T cells (ATCC number CRL-3216) were cultured in DMEM/High glucose (Hyclone, Logan, UT, USA), supplemented with 10% fetal bovine serum (PAN-Biotech, Aidenbach, Germany). The cells were maintained at 37 °C in an incubator (Thermo Fisher, Waltham, MA, USA) supplied with 5% CO_2_. The plasmids of pAC-Prkcaa-GFP and pAC-Prkcab-GFP were transfected into the 293T cells to characterize subcellular localization of zebrafish Prkcaa and Prkcab. The cells were seeded onto 35 mm glass bottom petri dishes at a density of 2 × 10^5^ cells per dish 1 day before transfection. Additionally, 1 h before transfection, the culture medium was replaced with fresh culture medium. The cells were transfected with 1 μg plasmid and 1 μL VigoFect reagent (Vigorous Biotech, Beijing, China). At 6 h after transfection, the old medium was replaced. At 24 h after transfection, the cells were washed twice with PBS and fixed with 4% PFA. After 3 washes with PBS, the cells were stained with DAPI-staining solution (BOSTER, Wuhan, China). Microphotography was conducted using a SP8 confocal microscope (Leica, Wetzlar, Germany) under a 63× oil immersion objective.

### 4.7. Generation of Gene-Knockout Zebrafish Lines

Gene-knockout zebrafish lines were generated using the CRISPR/Cas9 system according to our previous protocols [50]. The target sequence for the *prkcaa* gene was 5′-GGACCGCCGACTGGCTGTGG-3′, located in the 7th exon of the gene. The template for sgRNA was prepared by annealing the targeting oligo with the 80 nt chimeric sgRNA core sequence [51]. The sgRNA was generated using the TranscriptAid T7 High-Yield Transcription Kit (Thermo Fisher, Waltham, MA, USA). The purified sgRNA (final concentration 200 ng/µL) was mixed with Cas9 protein (final concentration 5 µM, EnGen Spy Cas9 NLS) from New England Biolabs (NEB, Ipswich, MA, USA). The sgRNA-Cas9 protein mixture was injected into single-cell embryos of WT zebrafish using a PICO-LITER syringe (WARNER, Holliston, MA, USA). The injection dose was 1–2 nL/embryo. To verify the effectiveness of the sgRNA, the injected embryos were lysed and used as templates for PCR. The sequence containing the target site was amplified using the primers gprkcaa-F: AGCCAATCGTCCTGTCATCTCT and gprkcaa-R: GGGTCATCTTCATCTGGAATGG. The PCR product was subjected to reannealing and T7 Endonuclease I digestion (T7E1, NEB, Ipswich, MA, USA). Positive results were determined by cleavage of the DNA fragment around the sgRNA target site. Successful editing of the target site was also confirmed by DNA sequencing.

The founders (F0) were cultured to adults and individually crossed with WT fish to generate F1 progenies. The F1 embryos of each F0 fish were subjected to T7E1 assay, as described above. The litters that passed the test were kept and raised to sexual maturation. The F1 fish were individually genotyped by T7E1 assay and DNA sequencing. The positive ones were crossed with WT fish to produce F2 offspring. The F2 fish were individually genotyped, and F3 offspring were generated by crossing the positive male and female with the same genotype. Finally, homozygous males and females of each genotype were selected from the corresponding F3 progenies and used to establish the homozygous gene-knockout fish lines.

### 4.8. Behavioral Assays

Fish behavioral assays were conducted using a ZebraTower system (Viewpoint Behaviour Technology, Lyon, France). The videos were analyzed using ZebraLab software (v3.22.3.31, Viewpoint Behaviour Technology, Lyon, France), Ethovision XT software (v14, Noldus Inc., Wageningen, The Netherlands) and Adobe Premiere Pro 2022 software (San Jose, CA, USA) to obtain the behavioral end points. To avoid effects of sex bias on the behavioral end points, only males were analyzed in this study. All the behavioral tests were performed between 11:00 a.m. and 6:00 p.m., except when otherwise stated. The temperature during the tests was maintained at 28 ± 1 °C. Behavioral recordings usually started after an acclimation period of 1–2 min. Videos were recorded at 25 fps and were pooled into 1 min time bins. Five behavioral tests were conducted in this study, including novel tank test, mirror biting test, social preference test, shoaling test and locomotion activity assay. Definition and significance of the behavioral end points are listed in Appendix A.

#### 4.8.1. Novel Tank Test

Novel tank tests were conducted to analyze anxiety-like behavior and exploratory activity of the experimental fish [52]. The fish were individually analyzed in a trapezoid tank (30 cm top/24 cm bottom × 10 cm × 15 cm) filled with 2.7 L of water (depth of the water was 10 cm). The behavioral trajectories were recorded for 20 min using a high-resolution camera on one of the trapezoidal sides of the tank. For the trajectory analysis, the novel tank was virtually divided into 2 vertical segments (top and bottom, 5 cm each). Behavioral indices, including total freezing time, time spent in the top, number of entries into the top, latency to the top, distance traveled in the top and average velocity, were obtained.

#### 4.8.2. Mirror Biting Test

Emotional activities such as sociality, aggressiveness and boldness of the fish were analyzed by mirror tests—as previously reported, with slight modifications [53]. Size of the mirror tank was 24 cm × 10 cm × 10 cm and depth of water in the tank was 2 cm. A mirror was placed on one of the square sides. The experimental fish was released at the center of the tank and videotaped for 15 min using a digital camera positioned right above the tank. For the behavioral analysis, the region within 1.5 cm from the mirror was regarded as mirror contact (mirror biting), and the region located between 1.5–5 cm from the mirror was defined as the approach area (AA). Behavioral indices, including number of mirror biting, mirror biting duration, number of entries into AA, time spent in AA, distance traveled in AA, total freezing time, total distance traveled and average speed, were obtained.

#### 4.8.3. Social Preference Test

Social preference tests were performed using a mating tank (48 cm × 8 cm × 10 cm) equally separated into 3 compartments (left, middle and right) by glass. Here, 5 zebrafish were put into the left chamber, while the right one was empty. The fish to be tested were placed into the center of the test chamber (middle). Video for each fish was recorded for 20 min. For data analysis, the middle chamber was virtually divided into two parts from the middle: the conspecific sector (CS) and the empty sector (ES). Social behavioral indices, including social preference value, ratio of time spent in CS and ratio of distance traveled in the CS, were calculated as indicated in Appendix A.

#### 4.8.4. Shoaling Test

A shallow tank (35 cm × 35 cm × 35 cm) that was filled with water to a depth of 2 cm was used for the shoaling tests. The fish were analyzed in groups, and each group contained 5 individuals. The fish were gently released into the center of the tank. Videos were recorded for 20 min for each group. The offline videos were analyzed by the ZebraLab multicount system to calculate shoal polarization; behavioral indices, including average interfish distance, average farthest/nearest distance and average velocity, were calculated using Ethovision XT 14 software. For shoal area and frequency of excursion measurements, Adobe Premiere Pro 2022 software was used to extract images from the videos (20 shots were obtained for each video, 1 per minute). The extracted images were further analyzed by ImageJ2 [54]. For each image, the distance between each fish was measured, and the center points of the fish were connected to form the shoal polygon. Existence of shoal was judged by the following threshold: at least 3 fish with the largest interfish distance of <12 cm. Excursion was identified when at least one fish demonstrated a distance >12 cm from its nearest neighbor.

#### 4.8.5. Locomotion Activity Assay

To assess the effects of the prkcaa mutation on circadian rhythmicity, locomotor activity of the fish was analyzed both in the morning (9:00–10:00 a.m.) and at night (9:00–10:00 p.m.). The fish were individually analyzed in the same tank used for the shoaling tests, and videos were recorded for 5 min. For recording the swimming trajectory at night, the fish were completely kept from light. The videos were analyzed to obtain the indices, including swimming speed and total traveled distance. Swimming activity of fish was scaled into 4 classes on the basis of speed, namely large (>15 cm/s), moderate (1–15 cm/s), small (0–1 cm/s) and freezing (no movement detected).

### 4.9. RNA Sequencing and Data Analysis

RNA sequencing (RNA-seq) was performed to analyze the effects of the *prkcaa* mutation on gene expression. Four-month-old males of the WT and mutants raised under the same condition were used for RNA-seq analysis. The fish were sampled both in the morning (9:00 a.m.) and at night (9:00 p.m.) to collect the brain tissue. Before tissue collection, the fish were anesthetized by immersion in ice water mixture for 3 min. The tissue collected from each individual was separately subjected to total RNA extraction and subsequent sequencing library construction and high-throughput sequencing. Moreover, 4 experimental groups were included in the assay, designated as WT_M (sample of the WT collected in the morning), WT_N (sample of the WT collected at night), 3145A_M (sample of the mutant collected in the morning) and 3145A_N (sample of the mutant collected at night). Each group contained 4 biological replicates; in total, 16 libraries were constructed and sequenced. Quality assessment of the RNA samples, library construction and high-throughput sequencing were conducted by Biomarker Technologies (Beijing, China). The libraries were sequenced for 150 bp from both ends (paired end) using an Illumina Novo6000 platform. The raw RNA-seq data of this study have been deposited in the NCBI Sequence Read Archive (SRA) under BioProject accession number PRJNA905701.

The workflow of RNA-seq data analysis was slightly modified from a previous study [50]. Briefly, the raw sequencing reads were preprocessed by fastp (v0.23.2) [55] with default settings to trim low-quality bases and filter low-quality reads. The clean reads were mapped to the genomic sequences of zebrafish (downloaded from Ensembl, http://asia.ensembl.org/, release-107, accessed on 20 December 2022) using hisat2 (v2.2.1) [56] with default parameters. The mapped reads were assigned to the gene features using featureCounts (subread-v2.0.3) [57]. Gene transcriptional abundance was calculated as fragments per kilobase per million mapped fragments (FPKM). Genes with FPKM > 0.1 in all the samples of at least 1 experimental group were regarded as expressed, and only the expressed genes were considered for the subsequent analyses. The genes differentially expressed between the experimental groups (Foldchange ≥ 1.5 and adjusted *p*-value ≤ 0.05) were identified using DESeq2 (v1.30.0) [58]. Principal component analysis (PCA) was performed using ArrayTrack [59]. GO (gene ontology), and KEGG (kyoto encyclopedia of genes and genomes) pathway enrichment analyses were performed using ClueGO (v2.5.9) [60]. Online software REVIGO [27] was used to reduce redundance of GO terms. GENE-E (https://software.broadinstitute.org/GENE-E, accessed on 20 December 2022) was used for cluster analysis and generation of heatmaps. Gene coexpression networks were constructed by the WGCNA R package [61], and hubs of the networks were identified by cytoHubba [62]. The networks for genes and GO terms were visualized and edited using Cytoscape (v3.9.1) [63]. All the resources used in this study are listed in Appendix A.

### 4.10. Statistical Analysis

Statistical tests were performed using SPSS statistics (v25.0, IBM Inc., Armonk, NY, USA) and R (v4.2.0). The data were usually expressed as mean ± standard error. Difference in the behavioral indices between the WT and mutant was analyzed by independent-samples *t* test. Difference in end points among multiple experimental groups was analyzed by one-way analysis of variance, followed by Tukey’s multiple comparison tests (*p* < 0.05). G-test using the R package RVAideMemoire was performed to test whether the prkcaa-mutation-affected genes were enriched for the circadian genes (genes differentially expressed between WT_N and WT_M).

## 5. Conclusions

Zebrafish Prkcaa and Prkcab share a high level of homology with mammalian Prkca. These two genes are extensively expressed in the zebrafish tissues, and the proteins are located mainly in the cytosol. The *prkcaa*-knockout zebrafish lines were generated and characterized. The loss of *prkcaa* had no impact on the development, growth and reproduction of zebrafish. Behavioral assays were performed to explore the functions of Prkcaa in regulating the social behavior of zebrafish. Prkcaa dysfunction resulted in anxiety-like behavior and impaired social preference in zebrafish. The mutation of *prkcaa* had a significant effect on the downregulation of a group of circadian genes at night, such as the circadian transcription factors *erg2a*, *fosaa*, *fosab* and *npas4a*. Furthermore, the mutants demonstrated inverse day–night locomotor activity, and they were more active at night than in the morning. Our data provide insights into the molecular functions of Prkca and link impaired social behavior with circadian rhythm disturbances.

## Figures and Tables

**Figure 1 ijms-24-03849-f001:**
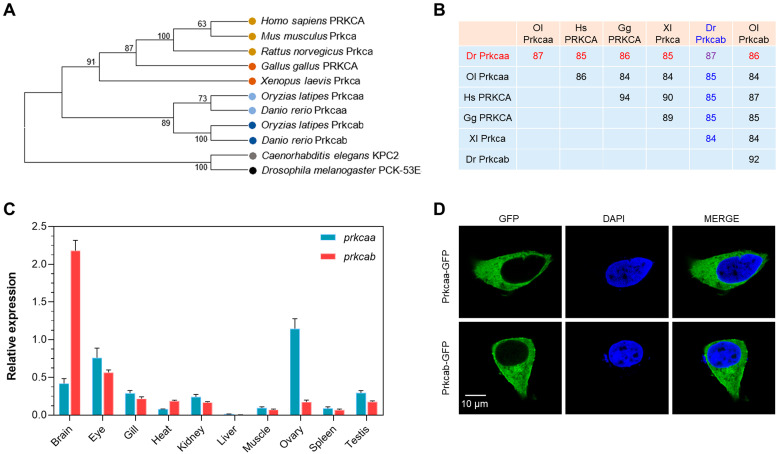
Molecular characterization of zebrafish Prkcaa and Prkcab. (**A**) A phylogenetic tree for PRKCA (or Prkcaa and Prkcab). The sequence accession numbers are listed in Appendix A. (**B**) Amino acid sequence identity between PRKCA (or Prkcaa and Prkcab). Dr, *Danio rerio*; Gg, *Gallus gallus*; Hs, *Homo sapiens*; Ol, *Oryzias latipes*; Xl, *Xenopus laevis*. Sequence identities between Dr Prkcaa and the others are shown in red, and those for Dr Prkcab are shown in blue. (**C**) Tissue expression patterns of zebrafish *prkcaa* and *prkcab*. The *y*-axis represents the expression of *prkcaa* and *prkcab* normalized to that of *18s rRNA*. The error bar stands for standard deviation of 3 technical replicates. (**D**) Subcellular localization of zebrafish Prkcaa and Prkcab expressed in the HEK293T cells.

**Figure 2 ijms-24-03849-f002:**
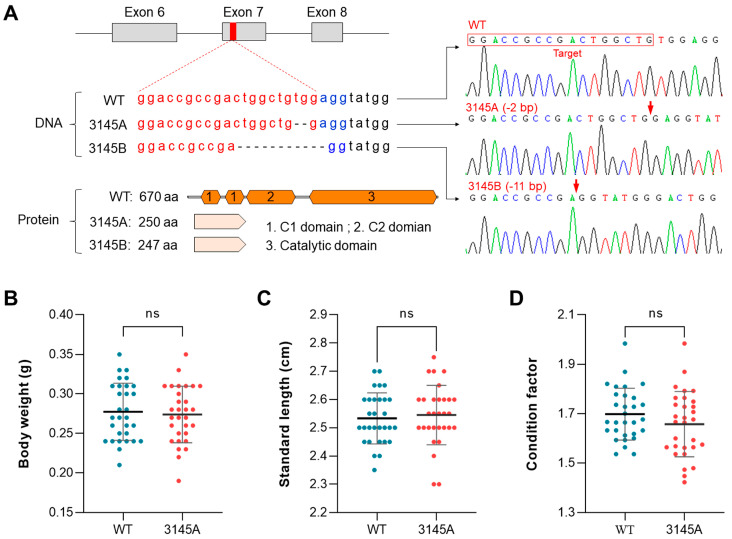
Genotypes and body measurements of the *prkcaa*-null zebrafish. (**A**) Genotypes of the *prkcaa*^−/−^ mutants. The target sequence was located in the seventh exon of the zebrafish *prkcaa* gene. Two mutant lines were generated and designated as ZKO3145A (3145A, −2 bp) and ZKO3145B (3145B, −11 bp). The target sequence, deleted nucleotides and the truncated proteins are illustrated. The red arrows indicate the joining sites of the broken DNA strands. (**B**–**D**) Measurements include body weight (**B**), standard length (**C**) and condition factor (**D**) for the adults of the WT and 3145A mutants. The error bars represent standard error (*n* = 30); ns, no significant difference. The different colored lines represent the four bases, A, T, C and G (shown above the lines). The different colored dots indicate the fish lines WT and 3145A (shown below the x-axis).

**Figure 3 ijms-24-03849-f003:**
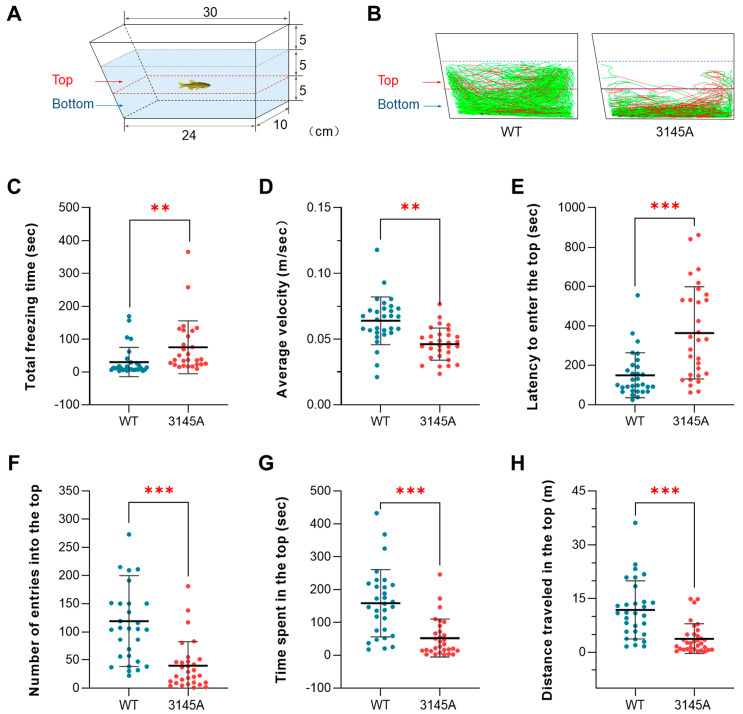
The *prkcaa*^−/−^ mutants demonstrated anxiety-like behavior. (**A**) A diagram for the novel tank. The tank was divided into 2 parts with a dashed red line: the top and the bottom. The numbers indicate tank size and water depth (cm). (**B**) Representative tracked trajectories of the WT and 3145A fish. The red and green line segments represent large (>15 cm/s) and moderate (1–15 cm/s) movements, respectively. (**C**) Total freezing time. (**D**) Average velocity. (**E**) Latency to enter the top. (**F**) Number of entries into the top. (**G**) Time spent in the top. (**H**) Distance traveled in the top. The error bars represent standard error (*n* = 30). **, *p* < 0.01; ***, *p* < 0.001.

**Figure 4 ijms-24-03849-f004:**
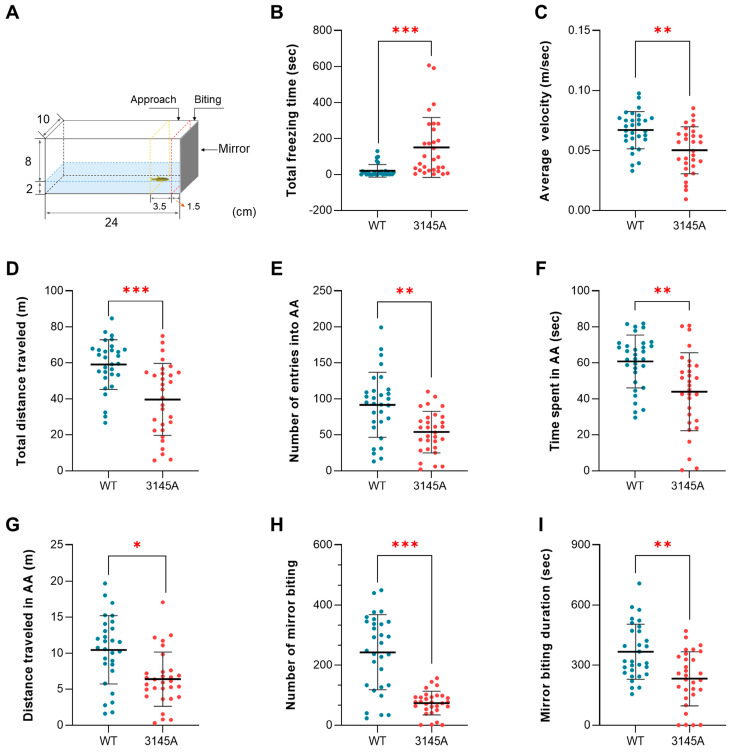
Loss of *prkcaa* decreased aggressiveness of zebrafish. (**A**) An illustration for the mirror biting test. The region adjacent to the mirror (1.5 cm from the mirror, dashed red line) is the mirror contact (mirror biting) area. That indicated by the dashed yellow line (1.5–5 cm from the mirror) is the approach area (AA). The numbers indicate the size of the aquarium (cm). (**B**) Total freezing time. (**C**) Average velocity. (**D**) Total distance traveled. (**E**) Number of entries into AA. (**F**) Time spent in AA. (**G**) Distance traveled in AA. (**H**) Number of mirror biting. (**I**) Mirror biting duration. The error bars represent standard error (*n* = 30). *, *p* < 0.05; **, *p* < 0.01; ***, *p* < 0.001; ns, no significant difference.

**Figure 5 ijms-24-03849-f005:**
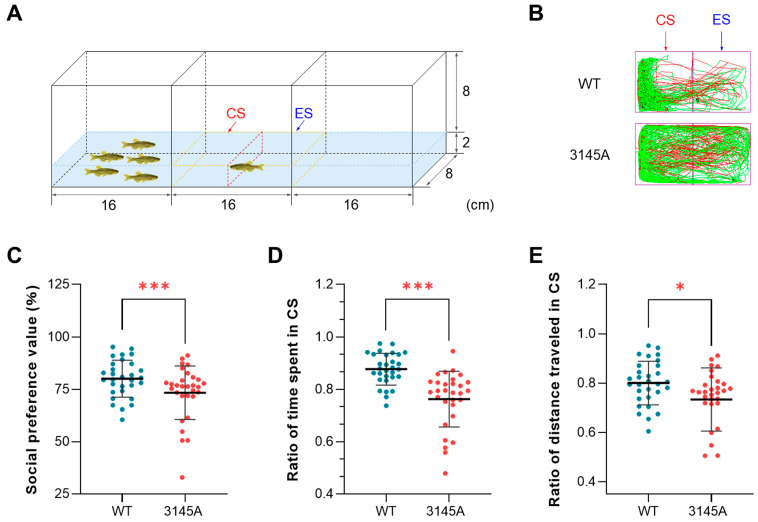
Dysfunction of Prkcaa attenuated the social preference of zebrafish. (**A**) A diagram for the social preference test. The region enclosed by the dashed yellow line is the video recording area, which is divided into the conspecific sector (CS) and the empty sector (ES) in the middle (the dashed red line). The numbers indicate the size of the tank (cm). (**B**) Representative tracked trajectories of the WT and 3145A fish. The red and green line segments represent large (> 15 cm/s) and moderate (1–15 cm/s) movements, respectively. (**C**) Social preference value. (**D**) Ratio of time spent in the CS. (**E**) Ratio of distance traveled in the CS. The error bars represent standard error (*n* = 30). *, *p* < 0.05; ***, *p* < 0.001.

**Figure 6 ijms-24-03849-f006:**
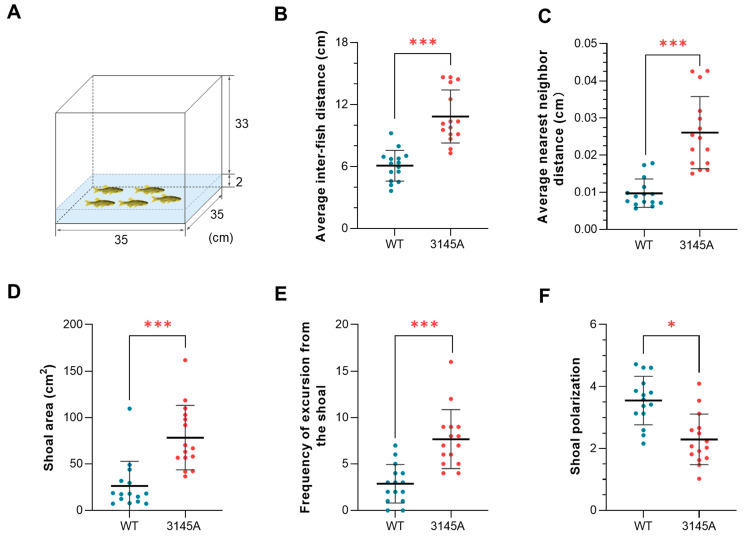
Deficiency in Prkcaa impaired the shoaling behavior of zebrafish. (**A**) An illustration of the shoaling test. The numbers indicate the size of the tank (cm). (**B**) Average interfish distance. (**C**) Average nearest neighbor distance. (**D**) Shoal area. (**E**) Frequency of excursion from the shoal. (**F**) Shoal polarization. The fish were analyzed in groups (each group included 5 individuals), and 15 groups were analyzed for each of the fish lines. The error bars represent standard error (*n* = 15). *, *p* < 0.05; ***, *p* < 0.001.

**Figure 7 ijms-24-03849-f007:**
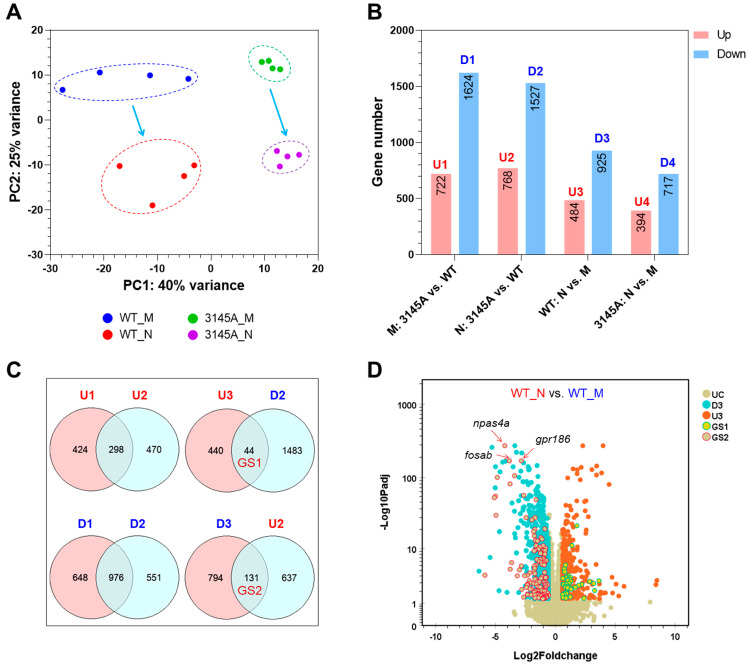
Effects of *prkcaa* mutation on gene expression. (**A**) Results of PCA. The proportion of variance explained by the first two PCs is displayed. (**B**) Numbers of the differentially expressed genes (DEGs). U1 to U4, the upregulated (up) gene sets; D1 to D4, the downregulated (down) gene sets. M and N represent the sampling time; M, morning (9:00 a.m.); N, night (9:00 p.m.). (**C**) Venn diagrams indicating the intersection between the gene sets. GS1 and GS2 represent the circadian genes regulated by the *prkcaa* mutation. (**D**) A volcano plot demonstrating the DEGs of the indicated gene sets. UC, unchanged; D3 and U3, the same as in (**B**); GS1 and GS3, the same as in (**C**). The representative circadian genes downregulated at night and upregulated by the *prkcaa* mutation are indicated.

**Figure 8 ijms-24-03849-f008:**
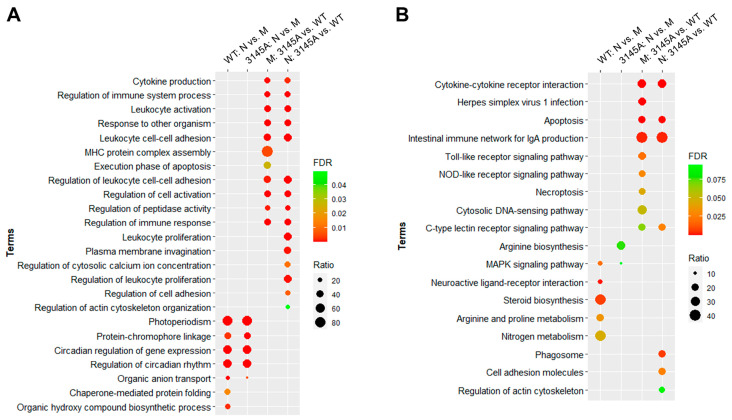
Top GO biological process and KEGG pathway enrichments for the DEGs. (**A**) Top GO biological process terms. (**B**) Top KEGG pathway terms. FDR, false discovery rate; ratio, proportion of the DEGs to all the genes associated with the corresponding functional term. Only terms with FDR < 0.05 are shown.

**Figure 9 ijms-24-03849-f009:**
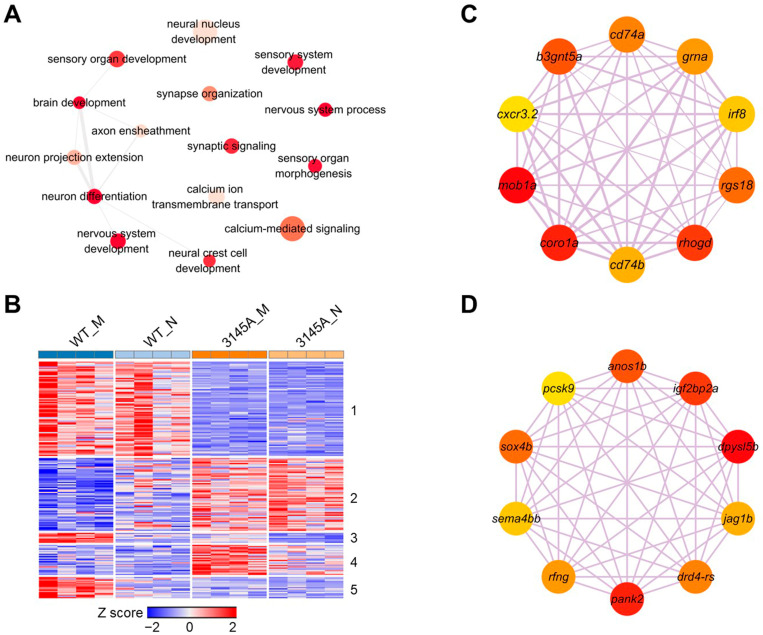
The genes affected by Prkcaa dysfunction and involved in neuron development and neural activity. (**A**) A network of the GO terms associated with neuron development and neural activity. The DEGs between WT and 3145A (both in morning and at night) were combined and subjected to GO enrichment analysis. Redundancy of the GO term list was removed by Revigo. Depth of color represents the *p*-value, and size of the nodes illustrates ratio of the DEGs associated with the corresponding GO term. (**B**) A heatmap demonstrating expression patterns of the genes affected by Prkcaa dysfunction and involved in neural functions. The genes were clustered into 5 clusters through the K-means method. (**C**) The Hub genes of cluster 1. (**D**) The Hub genes of cluster 2. Depth of color represents the *p*-value, and size of the nodes illustrates ratio of the DEGs associated with the corre-sponding GO term.

**Figure 10 ijms-24-03849-f010:**
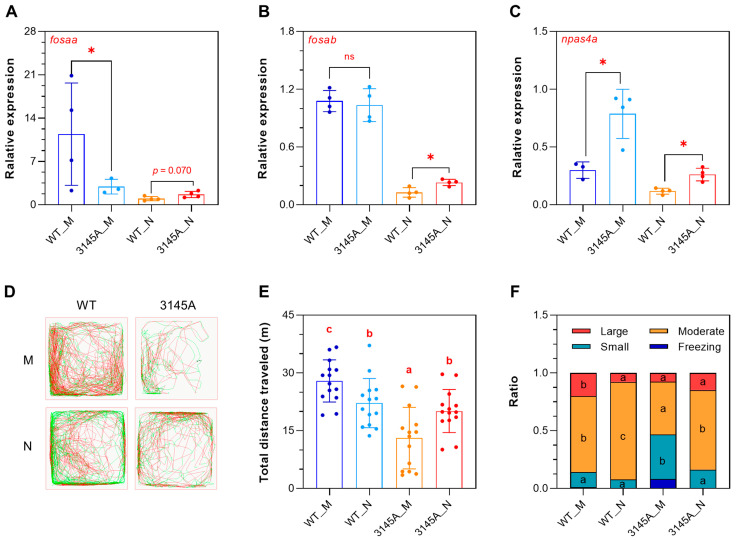
The *prkcaa* mutants are more active at night. (**A**–**C**) Expression of the representative circadian genes determined by qPCR. (**A**) *fosaa*, (**B**) *fosab*, (C) *npas4a*. (**D**) Representative tracked trajectories of the fish in the morning and at night. The red and green line segments represent large (>15 cm/s) and moderate (1–15 cm/s) movements, respectively. M, morning (9:00–11:00 a.m.); N, night (9:00–11:00 p.m.). (**E**) Total distance traveled. (**F**) Time ratio of the speed-based movement categories. The error bars represent standard error (*n* = 14). Different letters overlaid on the bar segments indicate significant difference between the means (* *p* < 0.05).

**Table 1 ijms-24-03849-t001:** A list of the behavioral assays and the purposes.

Experiment	Purpose
Novel tank test	To analyze anxiety-like behavior and exploratory activity of the experimental fish
Mirror biting test	To analyze emotional activities such as the sociality, aggressiveness and boldness of the experimental fish
Social preference test	To analyze the preference of the experimental fish to conspecifics
Shoaling test	To assess the social cohesion of the homogeneous group of the experimental fish
Locomotion activity assay	To assess the locomotion rhythmicity of the experimental fish

## Data Availability

The data presented in this study are available in the article and Appendix A.

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
