# Peer review of "Dysfunction of Prkcaa Links Social Behavior Defects with Disturbed Circadian Rhythm in Zebrafish"

_ijms, 2023, doi:10.3390/ijms24043849_

Round 1

Reviewer 1 Report

Nice work - but there are slight modifications as mentioned below

1. Reduce the size of the abstract.

2. Mention the research contributions and novelty of the work at end of introduction point by point.

3. Fig.1 is not clear and heading of the same figure is too much try to reduce for easy understanding - likewise for remaining figures as well.

4. What are the materials used in this experiment should be listed in a table for better understanding.

5. Methods should be explained in table or some other format for easy understanding.

6. Overall approach is good

Reviewer 2 Report

GENERAL COMMENTS:

The authors investigated the roles of Prkcaa in social behavior and circadian rhythm using prkcaa-deficient zebrafish. They utilized genome-editing methods and analyzed social behavior, using various situations in tanks, and circadian gene expression. The results suggested that PRKCAA is involved with regulation of social behavior and circadian rhythm. This manuscript contains lots of interesting points, but several issues need to be confirmed to conclude the present findings.

SPECIFIC COMMENTS:

1.      The authors demonstrated that deficiency of Prkcaa did not affect growth of zebrafish by measuring body weight, length and condition factor.  Did they analyze any other factors that represent physical and functional growth of the body, such as anatomical examination and blood test, for example?

2.      In line 160, the ones which had higher total freezing time and lower average velocity are mutants, not WT zebrafish?

3.      The authors found that Prkcaa had no effect on expression of core clock genes and explained that Prkcab may partially compensate the functions of Prckaa. If Prkcab cover the function of Prkcaa in 3145A mutants, how do they interpret the physiological meanings of the changes in morning-preferred genes?

Reviewer 3 Report

In this study, Hu et.al investigated how is Prkcaa deficiency affect the social behavior and circadian rhythm in zebrafish. Authors first generated this Prkcaa mutant zebrafish and showed that the mutant fish exhibited anxiety and aggressive behaviors and decreased social preference and shoaling behavior. Moreover, they also showed that he mutation of Prkcaa affected the circadian genes and locomotor activity. 

Overall, the manuscript is well written, and this study has an interesting topic, good rationale and clear hypothesis, detailed method section, discussion, and appropriate references. I only have several comments for clarification and further improvement. 

Comments:

1.     How specific is the Prkcaa knock out? Is there any compensatory effect from prkcab? 

2.     Figure 2D, how did you evaluate condition factor? A brief description should be included in the method section

3.     Are you able to measure the difference in Prkcaa protein size between the wild type and mutant?  What experiment did you do to confirm the generation of the mutant.

4.     How did you track the trajectories of the WT and 3145A fish in the novel tank test? Are the trajectories in figure 3B automatically generated or draw manually? 

5.     Did you also tried open field or a dark-light preference tests to evaluate anxiety behavior?

6.     Figure 6 is quite confusing with the n number, text wrote 5 fish/ group, but figure legend wrote n=15, this needs to be further clarified. 

7.     Figure 1B, what does the different colors mean? please described it in the figure legend or change it to one color if doesn’t mean anything. 

Round 2

Reviewer 2 Report

The authors responded to all of my questions thoroughly and added much information in the manuscript. The background of their research, the methods, the data and the discussion have become much clearer and more constructive.